# Low-Concentration Biological Sample Detection Using an Asymmetric Split Resonator Terahertz Metamaterial

**Yanchun Shen [1], Xiaoming Li [1], Jinlan Wang [1], Hongmei Liu [1], Junhao Jing [2], Xinxin Deng [3] and Dongshan Wei [3,*]**

[1] Information Engineering Institute, Guangzhou Railway Polytechnic, Guangzhou 510432, China
[2] Suzhou Ruizhixingyuan Intelligent Technology Co., Ltd., Suzhou 215301, China
[3] School of Electrical Engineering and Intelligentization, Dongguan University of Technology, Dongguan 523808, China
[*] Correspondence: dswei@dgut.edu.cn

**Abstract:** A simple and efficient THz metamaterial based on an asymmetric double split square array resonator was designed and fabricated. The sensitivity and Q value of the metamaterial sensor were theoretically analyzed to be 278 GHz/RIU and 11, respectively. Three typical biological samples with different concentrations were measured to validate the sensitivity of the THz metamaterial. For the bovine serum albumin (BSA) protein solution, the lowest detectable concentration (LDC) can reach to 1 μM, which is comparable to most recent reports. For the Staphylococcus aureus bacteria, the LDC is $1 \times 10^3$ cells/mL, and for the K-citrate solution, the LDC is 0.1 mM. Our studies indicate that the THz metamaterial may be effectively applied in the low-concentration detection of biological samples.

**Keywords:** terahertz spectroscopy; metamaterial; potassium citrate; lowest detectable concentration; bovine serum albumin; Staphylococcus aureus

## 1. Introduction

Recently, terahertz (THz) spectroscopy has gradually become an effective detection method widely applied in biological fields thanks to the inherent advantages of the THz wave, such as the good penetration, low photon energy, and pertinent energy levels for vibration and rotation of the molecular skeleton. In spite of these advantages, there are still challenges in accuracy and sensitivity for THz spectroscopy detection of biological samples because of the low power of THz source, the strong absorption of water in biological solutions, and the inhomogeneous scattering from the biological sample surface [1,2]. To overcome these challenges, high-sensitivity sensors and high-accuracy detection methods need to be urgently developed.

Metamaterials as a kind of periodic artificial composite material, have extraordinary physical properties, which can modulate the amplitude, phase, and polarization of electromagnetic waves and respond sensitively to the surrounding dielectric environment based on a surface-plasmon-polariton-like effect [3–6]. THz metamaterials can take advantage of the essential characteristics of both THz spectroscopy and metamaterials to acheive high sensitivity and low-concentration or even trace detection. THz metamaterials have been rapidly applied in many fields, including biomedicine [7–9], food safety [10,11], environment monitoring [12], material characterization [13], chemical and industrial product detection [14], and communication [15,16] since the original discovery by Yen et al. in 2004 [6]. For biological sample detection, since the resonant frequency of the THz material is closely related to its surrounding dielectric environment, once the analyte is placed on the surface of the THz metamaterial, the analyte concentration or content can be derived quantitatively according to the resonant frequency shift. As the detection process is simple and the detection result is accurate, THz metamaterials are widely used as sensitive sensors for low-concentration biological detection.

Researchers have designed and fabricated various THz metamaterial structures. Since the asymmetric split resonator (ASR) can break the symmetry of the metamaterial structure and excite the low-loss, high-Q, and polarization-dependent quadrupole and Fano resonances, ASR metamaterials have attracted a lot of attention [17,18]. In 2012, Al-Naib et al. [19] proposed a mirrored arrangement of ASRs that dramatically enhances the quality factor of the inductive-capacitive resonance. In 2014, Singh et al. [20] designed an array of ASR on a silicon substrate to excite a resonance with a high-Q factor and obtained a sensitivity of 36.7 GHz/RIU (RIU, refractive index unit). In 2015, Wen et al. [21] designed and fabricated a broadband THz metamaterial absorber using ASR and experimentally obtained a 0.82-THz bandwidth with absorptivity of more than 0.9, which is 3.4 times as wide as the 0.24-THz bandwidth of the symmetric dipole peak. Recently there have been a number of reports on the application of ASR array THz metamaterials to biological detection. Hu et al. [9] used an ASR metasensor of Al metal to realize real-time monitoring of the interaction between bovine serum albumin (BSA) and four drug molecules. Yan et al. [22] and Zhang et al. [23] proposed two novel biological sensors based on an EIT (electromagnetically induced transparency)-like metamaterial with the ASR arrays and utilized them to detect HSC3 dental cancer cells and glioma cells, respectively. Cheng et al. [24] designed an ASR metasensor of Cu metal for high-accuracy differentiation between A/G and A/G + IgG protein samples. To further improve the detection sensitivity, most recently, a number of THz metasurfaces and metalenses with asymmetric structures and different optical properties have also been designed and fabricated for modulating THz waves and detecting substances [25–28].

Taking the high-Q and low-loss performances of the ASR metamaterial into account, in this work we design and fabricate an asymmetric double-opening split square array metamaterial sensor and use it to detect three typical biological samples: bovine serum albumin (BSA) protein, Staphylococcus aureus (S. aureus) bacteria, and potassium citrate (K-citrate) solutions for validation of the sensitivity of the proposed THz metamaterial.

Due to its low cost, good water solubility, easy purification, and good biodegradability, BSA is very suitable for protein research. S. aureus is a human–animal pathogen widely present in the air environment, water, human and animal excreta, and can cause septic wound infections, pseudomembranous enteritis, and sepsis. Potassium citrate (K-citrate, $C_6H_5K_3O_7$), one kind of citrate salt, is not only one of the most commonly used food additives but also a therapeutic drug. Accurate detection of these biological samples is particularly important in practice.

The rest of this paper is organized as follows. In Section 2, an ASR metal array metamaterial is designed and fabricated, and the related performances of the metamaterial are discussed. In Section 3, the THz time-domain spectrometer (THz-TDS) setup, the biological sample preparation, and the THz spectroscopy measurements are introduced. In Section 4, the THz metamaterial sensing experiment results of these three biological samples are discussed and analyzed. In Section 5, conclusions of this work are given.

## 2. Design and Fabrication of the Asymmetric Split Square Resonator Metamaterial

The unit structure of the ASR square structure sensor is shown in Figure 1a. The sensor has a three-layer structure. The upper structured metal layer is gold with a conductivity of $4.09 \times 10^7$ S/m and a thickness of 200 nm. The lower layer is quartz with a thickness of 500 μm as the substrate. The intermediate layer uses an 8-nm-thick chromium as a binder. Electromagnetics simulations in CST Studio software were performed in a normalized TE incidence mode (electric field parallel to the opening direction of the metal square). The periodic boundary conditions were implemented along the x- and y-directions and the mesh size was automatically determined by the software. Figure 1b shows the transmission spectral profile of the sensor with two resonance frequency peaks at 1.025 THz and 1.535 THz with Q values of 30 and 11, respectively. The detailed structure geometry parameters of the sensor structure are summarized in Table 1.

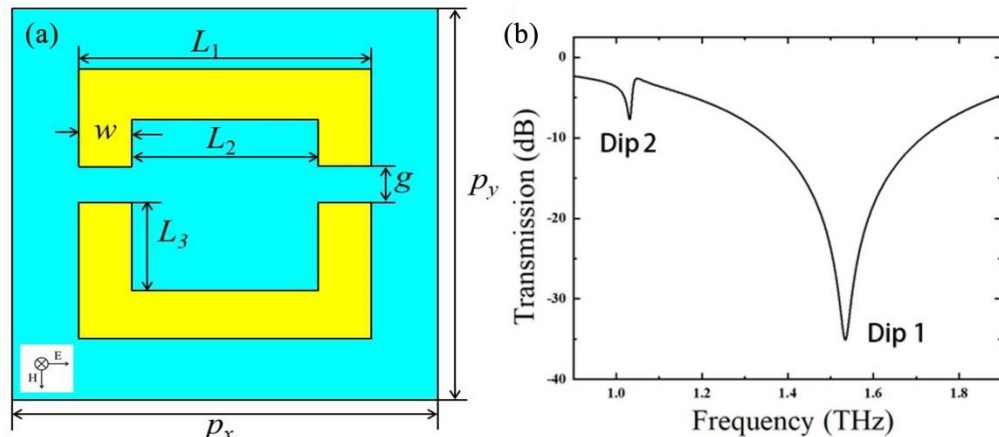

**Figure 1.** (**a**) Unit structure of the square ASR sensor; (**b**) Simulated transmission spectrum of the sensor.

**Table 1.** Summary of geometric parameters of the ASR sensors.

| Parameter Name | Parameter Setting |
| --- | --- |
| $L_1$ | 55 µm |
| $L_2$ | 35 µm |
| $L_3$ | 18.5 µm |
| g | 7 µm |
| w | 10 µm |
| $P_x$ | 80 µm |
| $P_y$ | 80 µm |

Although the Q value of the resonant peak at 1.025 THz (Dip 2) is high, the resonant frequency peak is not prominent and the resonant absorption at Dip 2 is much weaker compared to that at Dip 1. Therefore, the resonant frequency peak at Dip 1 was used to discuss the sensing performances of the sensor in the following. In order to explore the detection sensitivity of the ASR square structured metamaterials, we evaluated the sensing performances of this sensor by numerical simulations of the influences of the refractive index and the thickness of the analyte on the resonant frequency.

## 2.1. Influence of Refractive Index of the Analyte on Sensing Performance

We added a 20-µm-thick medium layer with variable refractive indices on the surface of the sensor to simulate the coverage of biological samples on the sensor. It is well known that the refractive indices of most biomolecular materials are between 1.2 and 2.0. Therefore, the range of the refractive index changed from $n = 1.2$ to $n = 2.0$ with an interval of 0.2, and the results obtained are shown in Figure 2a. The resonance peak of the terahertz metamaterial sensor had an obvious redshift when the analyte was added. The central resonance peak was also gradually red-shifted as the refractive index of the analyte increased. The redshift of the resonant peak, $\Delta f$, was 51 GHz when the refractive index was 1.2 and 277.5 GHz when the refractive index was 2.0. Variation of the extracted frequency shift data of the resonant peak with the refractive index is plotted in Figure 2b, and it can be seen that there is a linear relationship between the frequency shift and the refractive index. The linear fitting function corresponding to the resonant peak shift was $\Delta f = (278 \pm 6)\, n—(282 \pm 9)$ with $R^2 = 0.998$. The sensitivity of the simulated ASR sensor was determined to be 278 GHz/RIU, which is comparable to that of most proposed metamaterial sensor [20,24,29–31].

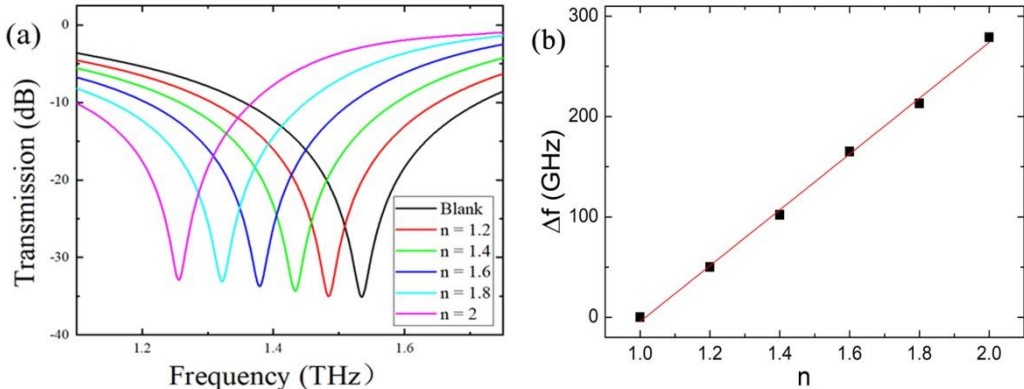

**Figure 2.** (**a**) Transmission spectra of the sensor covered by analyte with different refractive indices; (**b**) Plot of $\Delta f$ vs. *n* with the red line being the fitting line.

### 2.2. Influence of Thickness of the Analyte on Sensing Performance

We fixed the refractive index to be 1.4 and simulated the sensing performance of the sensor covered by the analyte with different thicknesses varying from 1 to 25 μm. Figure 3a shows the transmission spectra of the sensor covered by the analyte with different thicknesses. It can be seen that the resonant frequency shift $\Delta f$ of the resonance peak increased from 30 GHz to 105 GHz as the thickness *h* increased from 1 μm to 25 μm. Figure 3b clearly shows that $\Delta f$ increases exponentially with the increase in the thickness *h* and eventually saturates at about 15 μm, implying the electric field can extend up to 15 μm in the dielectric analyte, which is in good agreement with previous studies [20,29].

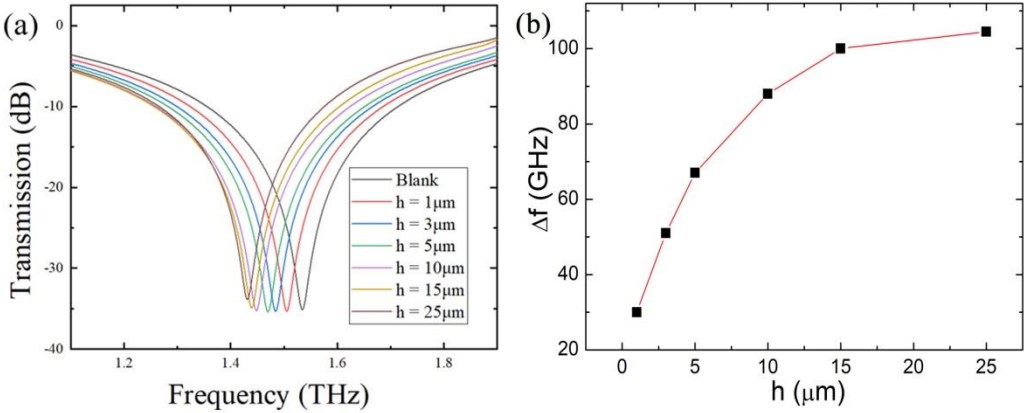

**Figure 3.** (**a**) Transmission spectra of the sensor covered by analyte with different thicknesses; (**b**) Plot of $\Delta f$ vs. *h* with the red curve included to guide the eyes.

### 2.3. Fabrication of the Metamaterial

The ASR array sensor was fabricated on the 500-μm-thick quartz crystal substrate using conventional photolithography. First, an 8-nm Cr layer was deposited on the substrates as a binder, then 200 nm of Au was deposited on the Cr via electron beam evaporation. Finally, lift-off technology was implemented to remove the metallic layer on the photoresist. A micrograph of the fabricated sensor is shown in Figure 4a with a square-shaped structure of 10 × 10 mm and a unit cell cycle of 80 × 80 μm. The fabricated sensor has good structural homogeneity with neat edges and no breakage under high-magnification microscopy.

After the fabrication of the ASR sensor was completed, the transmission spectrum of the blank sensor was first repetitively measured three times and compared with the simulation as shown in Figure 4b. From this figure, it can be seen that these two spectra have good consistency except that the experimental spectrum has some jitters at the two sides of the resonant frequency peak and has a wider frequency peak due to the fabrication

deviation, the environment noise, etc. The averaged measured resonant frequency of the blank metamaterial sensor is 1.556 THz with a standard deviation of 5 GHz and a lower Q value of 8.

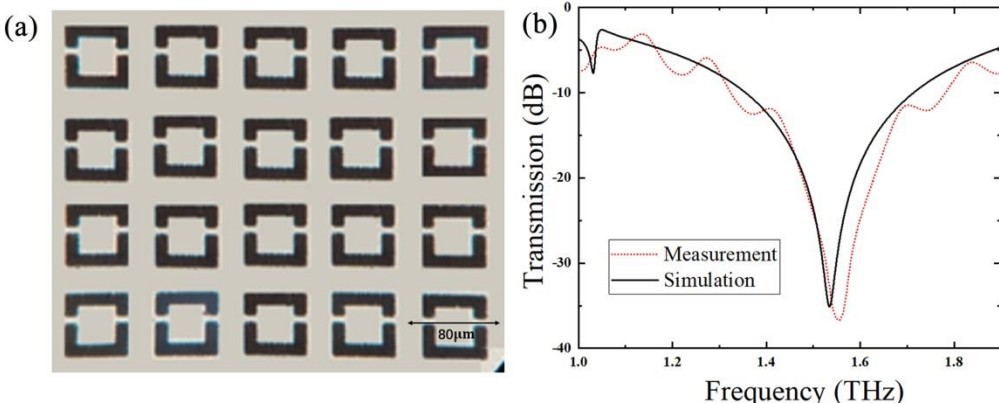

**Figure 4.** (**a**) Micrograph of the fabricated square ASR array metamaterial sensor; (**b**) THz transmission spectra of the sensor from the simulation and the experimental measurement.

## 3. Experiment Method and Sample Preparation

### 3.1. THz-TDS Spectrometer

A THz time-domain spectrometer (Advantest corporation, Tokyo, Japan Model TAS7500SU) was used in the experiment. During THz spectroscopy measurements, a maximal delay time of 132 ps was set and a frequency resolution of 7.6 GHz could be obtained. Under the setting, this spectrometer can stably produce a spectrum every 200 ms. The THz-TDS spectrometer is equipped with a dry air purge accessory with which the humidity of the detection chamber can be controlled around 5%. All terahertz spectroscopy measurements were performed in transmission mode at a temperature of $23.0 \pm 1.0\ ^\circ$C.

### 3.2. BSA, S. aureus and K-citrate Sample Preparation

All of these three biological samples used in the experiments were purchased from Shanghai Macklin Biochemical Technology Co., Ltd, shanghai, China. Bovine serum albumin (BSA) is a white to light yellow lyophilized powder with a molecular weight of 66.44 kDa and a solubility of 100 mg/mL. Staphylococcus aureus is a clear white solution with a concentration of $1 \times 10^6$ cells/mL and was stored at 2–8 $^\circ$C, protected from light, and refrigerated. For the BSA sample, 0.66 mg, 1.99 mg, 5.98 mg, and 9.97 mg of BSA powder were weighed on an electronic balance and dissolved in 10 mL of deionized water solution to obtain four different concentrations of 1, 3, 9, and 15 μmol/L (μM), respectively, in aqueous solution. For Staphylococcus aureus, the $1 \times 10^6$ cells/mL sample was diluted with deionized water at 1:9, 1:99. and 1:999 ratios to obtain $1 \times 10^5$, $1 \times 10^4$, and $1 \times 10^3$ cells/mL BSA solutions, respectively.

K-citrate has a solubility of 154 g/100 mL and a molecular weight of 306.39. To prepare K-citrate solutions at different concentrations, 1.54 g of K-citrate powder was weighed with an electronic balance in a 15-mL test tube, and 1 mL of deionized water was added into the test tube and then the test tube was shaken for 3 min to fully dissolve the K-citrate powder. An amount of 100 μL of K-citrate solution with a concentration of 5 mM was diluted to obtain another 3 concentrations of K-citrate solutions of 2.5, 1.0, and 0.1 mM.

Before THz spectroscopy measurements of the sensor covered by the biological solution, the blank sensor was soaked in anhydrous ethanol for half an hour and completely purged with nitrogen gas. Each biological sample was dropped on the surface of the sensor using the pipetting gun and a homogenizer was used to ensure that the sample was evenly spread on the surface, followed by drying in a drying oven at 40 $^\circ$C for about 5 h. Since the solution volume of the sample dropped on the surface of the sensor was fixed, the thickness of samples at different concentrations was the same. When the solution sample

was dried on the surface of the sensor, the biological solute would distribute evenly at the surface of the sensor because the concentration of the sample is not high and overlap or superposition of solute particles will not happen. Therefore, even in the dried state, the thickness of samples with different concentrations on the surface of the sensor also was the same.

## 4. Result and Discussion

### 4.1. THz Spectroscopy Measurements of Different Concentrations of BSA Solution

Transmission spectra of the blank THz metamaterial sensor and the THz metamaterial sensor covered by BSA samples at different concentrations were measured and the results are shown in Figure 5a. After the BSA sample was dropped and dried on the sensor, the resonant frequency of the sensor had a prominent redshift. With the increase in the BSA concentration, the redshift of the resonant peak becomes larger. This is because the increased concentration of BSA sample gives rise to a higher equivalent capacitance, which will lead to the decrease in the resonant frequency according to principles of the LC circuit [32,33]. When the BSA concentration was lowest at 1 μM, the resonance peak frequency at 1.548 THz was obviously redshifted by 8 GHz relative to the resonance frequency of the blank sensor itself, comparable to the frequency resolution of 7.6 GHz, indicating the THz metamaterial sensor can detect a BSA sample at a concentration of 1 μM, which is defined as the lowest detectable concentration (LDC) of the sensor and is comparable to the latest results of 1.52 μM [7,8], 0.53 μM [34], and 0.15 μM [35]. To quantify the relation between the resonant frequency redshift $\Delta f$ and the concentration $C$, Figure 6b was plotted. It can be seen there is an approximate linear relation between $\Delta f$ and $C$ with an equation of y = 6.56 + 2.18 C, illustrating a sensitivity of 2.18 GHz/μM for BSA detection. We explain the linear relationship between $\Delta f$ and $C$ at the present low concentration range as follows. Since the frequency shift is mainly dependent on the resonance of the sensor and the solute particles at the openings of the sensor, when the concentration is low, the number of the dispersed solute particles at the openings increases in proportion with the increase in the concentration, which will give rise to a linear relationship between $\Delta f$ and $C$. It can be predicted that when the concentration of the BSA solution continues to increase, the linear relationship will not remain since the solute particles at the openings of the sensor will probably aggregate and when the concentration increases to a critical value, all of the space at the openings of the sensor is fully occupied by solute particles and the resonant frequency shift will not change.

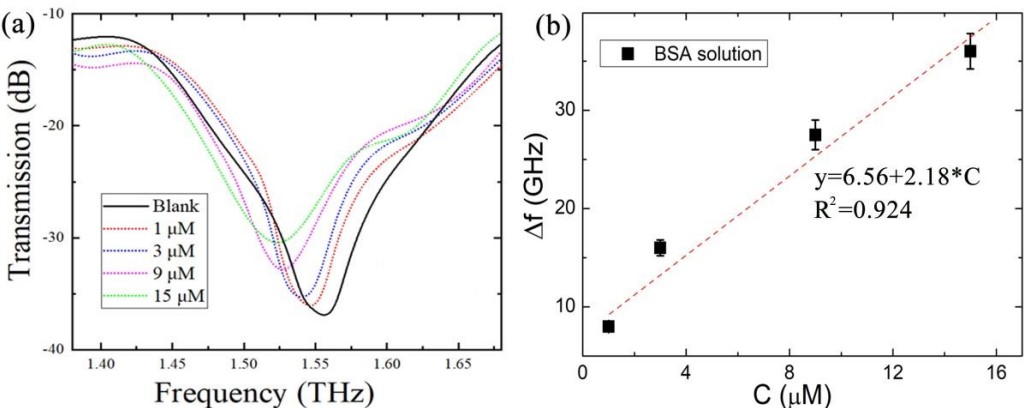

**Figure 5.** (**a**) Transmission spectra of the blank sensor and sensor covered by BSA sample at different concentrations. (**b**) Relationship between $\Delta f$ and the BSA concentration with the dash line being the fitting line.

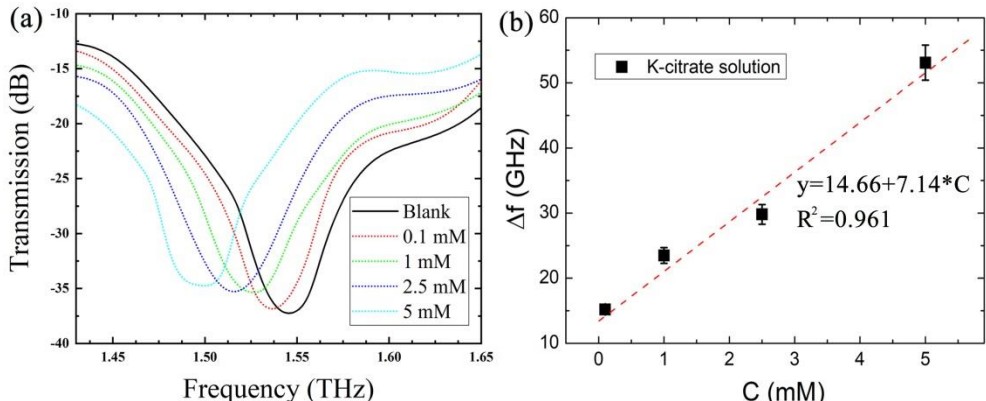

**Figure 6.** (**a**) Transmission spectra of the blank sensor and sensor covered by K-citrate sample at different concentrations. (**b**) Relationship between $\Delta f$ and the K-citrate concentration with the dash line being the fitting line.

In addition, considering the frequency resolution of the THz-TDS spectrometer and the sensitivity of the sensor for BSA solution, the uncertainty of the detected concentration of BSA was estimated to be 3.7 µM, which is comparable to the LDC of BSA.

*4.2. THz Spectroscopy Measurements of Different Concentrations of K-citrate Solution*

Figure 6a shows the transmission spectra of the blank metamaterial and the K-citrate solution with different concentrations varying from 0.1, 1, 2.5, to 5 mM. It can be clearly seen that as the concentration of K-citrate solution increased, the position of the resonance peak of the transmission spectrum was red-shifted. The resonant frequency redshifts caused by the concentration increase in K-citrate were 15, 24, 30, and 53 GHz, respectively, as shown in Figure 6b. The lowest detectable concentration that could be measured by the sensor was 0.1 mM, and the fitting equation is $y = 14.66 + 7.14 \, C$ with $R^2$ of 0.961, indicating there is a good linear relation between the resonant peak frequency redshift and the concentration. The experimental results show that the fabricated ASR sensor is suitable for distinguishing K-citrate solutions at different concentrations. Similarly, according to the detection sensitivity of 7.14 GHz/mM, the uncertainty of the detected concentration of K-citrate was estimated to be 1.1 mM, which is much larger than the LDC due to the lower detection sensitivity compared with that of the BSA sample.

*4.3. THz Spectroscopy Measurements of Different Concentrations of S. aureus*

Using the same experimental conditions, the THz transmission spectra of a S. aureus sample at different concentrations were measured as shown in Figure 7a. Similarly, when the S. aureus sample was dropped on the sensor, the resonant frequency of the sensor had a prominent redshift. With the increase in the S. aureus concentration, the redshift of the resonant peak became larger. Figure 7b shows the quantitative relation between $\Delta f$ and the concentration and a logarithmic fitting equation of $y = 10.28 + 14.65 \, \log C$ with $R^2$ of 0.979 can be obtained. It is worth noting that for BSA and K-citrate solution samples, the frequency shifts have a linear dependence on their concentrations, while for S. aureus solution, the frequency shift has a logarithmic dependence on the concentration. The reason may be that S. aureus has a large size comparable to the openings of the sensor and it can easily and fully occupy the space of all openings of the sensor at the present concentration range. S. aureus out of the openings of the sensor will contribute little to the frequency shift. Therefore, the frequency shift will not increase in proportion with the concentration, but in a slow logarithmic way. We also note that at the concentration of $1 \times 10^3$ cells/mL, the resonant peak had a redshift of 55 GHz, much larger than the frequency resolution of the spectrometer, indicating the lowest detectable concentration of the sensor can easily reach to $1 \times 10^3$ cells/mL. Similarly, the uncertainty of the detected concentration of S.

aureus was estimated to be $0.55 \times 10^3$ cells/mL, which is in good agreement with the lowest detectable concentration.

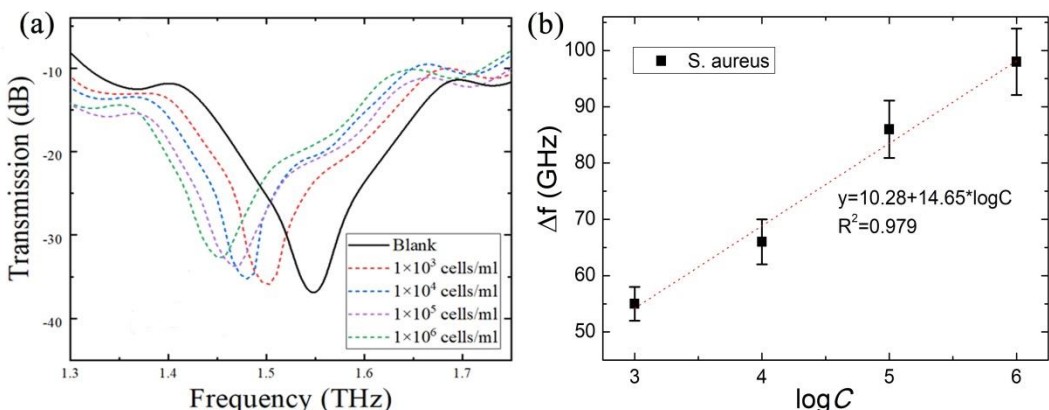

**Figure 7.** (**a**) Transmission spectra of the blank sensor and sensor covered by S. aureus sample at different concentrations. (**b**) Plot of $\Delta f$ versus log$C$ with the dash line being the fitting line.

## 5. Conclusions

In summary, we proposed a simple and efficient THz metamaterial based on an asymmetric split square array structure. The sensing performances of the metamaterial varying with the refractive index and the thickness of the analyte were analyzed by simulations. The theoretical Q value and sensitivity of the metamaterial sensor can reach to 11 and 278 GHz/RIU, respectively. With the designed and fabricated THz metamaterial, three typical biological solution samples with different concentrations were measured. For the BSA protein, the lowest detectable concentration can reach to 1 uM; for the S. aureus bacteria, the LDC is $1 \times 10^3$ cells/mL; for the K-citrate solution, the LDC can reach to 0.1 mM. Our studies indicate that the THz metamaterial may be effectively applied in low-concentration detection of biological samples including biomolecules, cells, and bacteria. Although the present LDCs still deviate from the concentration of biological samples in practical utilization, with the development of the THz metamaterial sensing technology, sensors with better stability and higher sensitivity will be designed and fabricated and the specific methods to remove the disturbance from other substances and the environment noise will be presented. Eventually, trace detection of biological samples at the concentration of nM using THz spectroscopy will be realized.

**Author Contributions:** Conceptualization, Y.S. and D.W.; methodology, X.D., X.L. and J.W.; software, H.L.; validation, D.W.; formal analysis, Y.S. and X.D.; investigation, Y.S. and X.L.; resources, J.J. and D.W.; data curation, J.W.; writing—original draft preparation, Y.S.; writing—review and editing, D.W.; visualization, Y.S.; supervision, D.W.; project administration, D.W.; funding acquisition, Y.S. and D.W. All authors have read and agreed to the published version of the manuscript.

**Funding:** This work has been partially supported by Guangzhou Science and Technology Project (Grant 202102080473) and Guangdong Basic and Applied Basic Research Foundation (2021B1515140018, 2021A1515110664).

**Institutional Review Board Statement:** Not applicable.

**Informed Consent Statement:** Not applicable.

**Data Availability Statement:** Data underlying the results presented in this paper are not publicly available at this time but may be obtained from the authors upon reasonable request.

**Conflicts of Interest:** There are no conflicts of interest to declare.

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
