# Peer review of "Low-Concentration Biological Sample Detection Using an Asymmetric Split Resonator Terahertz Metamaterial"

_photonics, doi:10.3390/photonics10020111_

Round 1

Reviewer 1 Report

The authors describe the design, fabrication and test of a simple and efficient THz metamaterial based on an asymmetric double split square array resonator for the detection of low-concentration biological samples.

The paper is well written: materials, methods and results are clearly and concisely described.

The biological samples used to test the Asymmetric Split Resonator are significant and the shift of resonant frequency vs. sample concentration is clearly reported in the text and in the figures.

The achieved limit of detection gives hope that the further development of THz metamaterial sensing technology will allow trace detection of biological samples at the concentration of nM or pM.

I suggest the following adjustments/corrections:

Lines 23-24 should read “Terahertz (THz) spectroscopy has gradually become an effective detection method widely applied in biological fields thanks to the inherent advantages of the THz wave like…”

Lines 58-59 should read “Recently there have been a number of reports on the application of THz metamaterials ASR arrays to biological detection”

Line 114 – check the units “The lower layer is quartz with a thickness of 500 mm as the substrate”

Line 160 – check the units “The ASR array sensor was fabricated on the 500 mm thick quartz crystal substrate”

The paper is recommended for publication provided the above minor corrections are made.

Reviewer 2 Report

The paper studies an asymmetric split resonator (ASR) THz metamaterial and demonstrates its applicability for detecting low concentrations of three types of biological samples. The general idea of using metamaterials with narrow resonance lines, which are very sensitive to environmental parameters, is now at the forefront of research in the detection of substances. So, the subject of the paper is quite relevant and in demand.

The ASR-based metasensors have already shown their fine application possibilities in a number of works including the ones sited in the presented paper. In continuation of previous studies, the authors have designed a new variant of an ASR metasensor and applied it to 2 new biological samples, Staphylococcus aureus (S. aureus) bacteria and potassium 69 citrate (K-citrate), and to 1 sample which has been probed with other sensors, Bovine serum albumin (BSA). The result on the concentration limit of detection (1  μΜ) obtained for BSA is shown to be comparable or somewhat worse than that obtained with other sensors.  So, the main new result of the research is that the proposed metasensor structure is sensitive to low concentrations of these 3 samples if they are prepared in the same way as the paper describes.

However, a number of issues need to be clarified:

1. How the thickness or porosity of the samples can influence the final conclusion about the concentration level when only the shift of the transmission minimum is measured? The numerical calculations made for different thicknesses of the samples maybe can help in answering on this question, but they are not analyzed properly in the paper.

2. At what conditions this sensor can be used for measuring the concentration quantity? What is the final uncertainty of the concentration value detected for each substance?

3. In what real application tasks can this sensor be used - as for example? What are the requirements for the practical conditions of its use (parallel substances, method of application, and others)?

 A few points regarding text checking:

Lines 154-155 “..and a change of 1 μm can causes  an obvious change of the resonant frequency. 

Line 252 “..fabricated ASR sensor with is suitable for..”

Reviewer 4 Report

Overall, there are so many problems with the paper that I didn't read it to the end. There are some major issues that have to be resolved before a publication can be considered.

Please see the comments in the attachments.

Round 2

Reviewer 2 Report

The authors answered most of the questions in their response letter. Nevertheless, the adjustments made to the text of the paper are still insufficient:

1. The question of why a change in concentration should lead to a purely linear shift in the resonant frequency has remained insufficiently clarified in the text. According to theoretical calculations, the frequency shift is influenced by 2 parameters, the thickness and the refractive index. Both parameters could vary with changes in concentration in the experiment. But fundamentally, a change in thickness in a certain range leads to a nonlinear change in frequency. How did the thicknesses of the samples change when the concentrations changed in the experiment? It is necessary to describe in the text. And to explain further why the linear approximation of experimental dependencies is correct.

 2. The authors cite in the answer (but not in the revised paper!) a useful analysis of uncertainties of the final concentration values that can be determined for each substance: "..uncertainty of the final concentrations to be 2.3mM, 0.34×103 cells/mL and 0.7 mM for BSA, S. aureus and K-citrate samples, respectively". I believe these values should be presented in the paper. And it will be useful to compare them with the final results on LoDs.

Author Response

please the attachment

Round 3

Reviewer 2 Report

Please, correct the item number: "3.2. BSA, S. Aureus ..".

Reviewer 4 Report

Much better than the first version.